# Light Exposure, Physical Activity, and Indigeneity Modulate Seasonal Variation in NR1D1 (REV-ERBα) Expression

**DOI:** 10.3390/biology14030231

**Published:** 2025-02-25

**Authors:** Denis Gubin, Sergey Kolomeichuk, Konstantin Danilenko, Oliver Stefani, Alexander Markov, Ivan Petrov, Kirill Voronin, Marina Mezhakova, Mikhail Borisenkov, Aislu Shigabaeva, Julia Boldyreva, Julianna Petrova, Dietmar Weinert, Germaine Cornelissen

**Affiliations:** 1Department of Biology, Tyumen Medical University, 625023 Tyumen, Russia; 2Laboratory for Chronobiology and Chronomedicine, Research Institute of Biomedicine and Biomedical Technologies, Tyumen Medical University, 625023 Tyumen, Russia; kvdani@mail.ru (K.D.); h_aislu@mail.ru (A.S.); tgma.06@mail.ru (J.B.); 3Tyumen Cardiology Research Centre, Tomsk National Research Medical Center, Russian Academy of Science, 119991 Tyumen, Russia; 4Laboratory for Genomics, Proteomics, and Metabolomics, Research Institute of Biomedicine and Biomedical Technologies, Medical University, 625023 Tyumen, Russia; sergey_kolomeychuk@rambler.ru (S.K.); alexdoktor@inbox.ru (A.M.); lirik92@list.ru (K.V.); chiz.maslova@yandex.ru (M.M.); 5Laboratory of Genetics, Institute of Biology of the Karelian Science Center of the Russian Academy of Sciences, 185910 Petrozavodsk, Russia; 6Institute of Neurosciences and Medicine, 630117 Novosibirsk, Russia; 7Department Engineering and Architecture, Institute of Building Technology and Energy, Lucerne University of Applied Sciences and Arts, 6048 Horw, Switzerland; oliver.stefani@hslu.ch; 8Department of Biological & Medical Physics UNESCO, Medical University, 625023 Tyumen, Russia; petrovtokb@mail.ru (I.P.); pimtmn@mail.ru (J.P.); 9Department of Molecular Immunology and Biotechnology, Institute of Physiology of the Federal Research Centre Komi Science Centre of the Ural Branch of the Russian Academy of Sciences, 167982 Syktyvkar, Russia; borisenkov@physiol.komisc.ru; 10Institute of Biology/Zoology, Martin Luther University Halle-Wittenberg, 06120 Halle, Germany; dietmar.weinert@zoologie.uni-halle.de; 11Department of Integrated Biology and Physiology, University of Minnesota, Minneapolis, MN 55455, USA; corne001@umn.edu

**Keywords:** light, circadian, clock genes, gene expression, NR1D1, REV-ERB, actigraphy, Arctic, seasons

## Abstract

We found that the morning nuclear receptor subfamily 1 group D member 1 (NR1D1 or REV-ERBα) expression in the human mononuclear cells of Arctic residents has a seasonal rhythm, with its peak at the summer solstice and nadir at the winter solstice. Actigraphy data indicated that the expression of REV-ERBα was significantly associated with physical activity and natural daylight exposure, suggesting a responsiveness to external cues. We also found that indigenous populations have elevated REV-ERBα expression compared to non-indigenous populations, particularly at the summer solstice. These results highlight the dynamic regulation of REV-ERBα and its sensitivity to both environmental and genetic influences. These findings suggest the potential roles of REV-ERBα in seasonal adaptation and/or metabolic differences among human populations and encourage further exploration of these findings, which may shed light on the complex interplay between environmental cues, genetic backgrounds, and REV-ERBα-mediated human physiological processes.

## 1. Introduction

Nuclear receptor subfamily 1 group D members 1 and 2 (NR1D1/NR1D2), also known as REV-ERBα and REV-ERBβ, were named for their unique genomic organization, encoded by the opposite DNA strand of the ERBA gene, which encodes the thyroid hormone receptor-α. Unlike many nuclear receptors that function as obligate heterodimers, REV-ERBs typically operate as monomers recognizing a specific half-site sequence [1]. REV-ERBα and REV-ERBβ act as transcriptional repressors through two distinct mechanisms: they compete with RORs to bind to RORE-containing enhancers and recruit NCoR complexes that include the epigenomic modulator histone deacetylase 3 (HDAC3) to actively repress transcription [2]. Both receptors exhibit overlapping expression patterns and circadian rhythms, highlighting their significant and coordinated roles in regulating transcription. The genes encoding both REV-ERBs exhibit strong circadian rhythmicity in various mammalian tissues [2,3,4], with peak expression during the light phase in laboratory animals [3,5]. They are expected to peak early in the morning with a nadir in the evening in human blood cells [6,7]. REV-ERBs modulate the rhythmic expression of several core clock genes, including Bmal1, Npas2, Clock, and Cry1, as reviewed in [2,8]. Knocking down both REV-ERBs profoundly alters the circadian free-running period and also impairs lipid metabolism [3,9]. Beyond their role in the core clock loop, REV-ERBs directly regulate circadian-controlled genes (CCGs), facilitating the tissue-specific modulation of the circadian transcriptome, as reviewed in [2,8]. Rev-erbα abundance is regulated by the molecular clock, but it can be influenced by environmental factors like light and food, depending on the tissue. Rev-erbα’s activity can also be affected by its ligand (heme), circadian fluctuations, and conditions that alter its levels, such as cold exposure and glucocorticoids [10]. Consequently, Rev-erbα represses genes in a tissue-specific manner, resulting in distinct functions across different cell types.

Circadian disruption, such as that experienced in shift work, can lead to metabolic diseases due to a mismatch between internal biological clocks and environmental cues. While the SCN REV-ERB nuclear receptors are not essential for maintaining rhythmicity, they do influence the length of the free-running period. This property is illustrated by studies showing that mice lacking these receptors exhibited shortened rhythms and increased weight gain on an obesogenic diet, a condition that improved when environmental lighting was aligned with their altered 21 h clock [11]. The whole-brain knockout of REV-ERBα affects not only the circadian locomotor activity rhythm, but can even be essential for the circadian rhythmicity of food intake and food-anticipatory behavior [12]. Furthermore, REV-ERBα may mediate light-dependent changes in mood via serotonin modulation [13,14]. The mononuclear expression of Rev-erbs is also integrated into a BodyTime assay that aims to determine the internal circadian time in humans using a single morning blood sample [15].

However, the expression of REV-ERBα and REV-ERBβ has been much less studied in humans compared to experimental laboratory rodents, which are primarily nocturnal, in contrast to diurnal humans. It remains largely unknown whether the expression of REV-ERBs is influenced by the challenging environmental light conditions at high Arctic latitudes and whether differences exist between adapted native populations of these regions and non-native newcomers. Therefore, we aimed to evaluate the expression of the major REV-ERBα by analyzing the relative abundance of Rev-erbα mRNA in blood mononuclear cells from both Arctic native and non-native residents during equinoxes and solstices, as part of the broader Light Arctic project.

## 2. Materials and Methods

This study adhered to the tenets of the Declaration of Helsinki and was approved by the Ethics Committee of Tyumen State Medical University (Protocol No. 101, 13 September 2021). Written informed consent was obtained from all participants, as published previously [16,17].

### 2.1. Subjects and Data Collection

Situated at latitude 65°58′–66°53 N and longitude 66°60′–76°63′ E, 29 Arctic residents (age range 18–52, mean age 39.3 years, 82.8% women, 27.6% Arctic natives) provided seven-day actigraphy records and morning blood samples in each season during the spring/autumn equinoxes and winter/summer solstices, as previously described [16,17].

### 2.2. NR1D1 (REV-ERBα) Expression

All blood samples were collected at 08:00 during the weekends closest to the winter and summer solstices and to the spring and autumn equinoxes, as described elsewhere [16,17]. To keep the genetic material intact during transportation, the RNA in patient blood samples was preserved using Blood RNA Stabilizer reagent from Inogen (St. Petersburg, Russia). Total RNA was isolated using PureZol reagent (Bio-Rad, Hercules, CA, USA) according to the manufacturer’s protocol. The RNA concentration and integrity were controlled via denaturing gel electrophoresis in 2.5% agarose. Synthesis of the first strand of cDNA was performed using the MMLV Kit from Eurogen (Moscow, Russia) according to the company’s instructions. At least 100 ng of total RNA was used in the cDNA synthesis reaction. NR1D1 (REV-ERBα) expression was determined by real-time PCR on a CFX96 Nucleic Acid Amplification Thermal Cycler (Bio-Rad Laboratories Inc., Hercules, CA, USA). The glyceraldehyde-3-phosphate dehydrogenase (GAPDH) gene was used as a reference gene. The primer sequences for real-time PCR were as follows: NR1D1 forward 5′-CCATCGTCCGCATCAATCAATCG-3′; NR1D1 reverse 5′-GCATCTCAGCAAGCAGCATCCG-3′; GAPDH forward 5′-GTCTCCTCTGACTTCAACAGCG-3′; and GAPDH reverse 5′-ACCACCCTGTTGCTGTAGCCAA-3′. Data from the qPCR results were analyzed by the CFX Manager 3.1 software (Bio-Rad, Hercules, CA, USA). The relative level of gene expression was calculated using the 2^−ΔΔCq^ formula [18].

### 2.3. Actigraphy

A seven-day actigraphy protocol using the ActTrust 2 device (Condor Instruments, São Paulo, Brazil) followed the methodology outlined in previous studies [16,17,19,20]. ActTrust 2 recorded various metrics at one-minute intervals, including motor activity (Proportional Integrative Mode, PIM), wrist skin temperature (wT), and light intensity (lux). Additionally, it measured infrared, red, green, blue, and ultraviolet A (UVA) and B (UVB) light intensities (in μw/cm^2^). Parametric endpoints such as MESOR (M), 24 h amplitude (24h-A), and acrophase were computed for PIM, wT, light exposure (LE), and blue light exposure (BLE). Non-parametric endpoints such as activity during the peak 10 h period (M10), M10 onset, the least active 5 h period (L5), L5 onset, inter-daily stability (IS), intra-daily variability (IV), relative amplitude (RA = (M10 − L5)/(M10 + L5)), and the circadian function index (CFI) were estimated for PIM and BLE using the ActStudio software 1.0.25.(Condor Instruments, São Paulo, Brazil). Sleep parameters such as bedtime, wake time, time in bed, total sleep time, sleep efficiency, sleep latency, and wake after sleep onset (WASO) were calculated along with their standard deviations (SD) using the same software.

### 2.4. Data and Statistical Analyses

Statistical analyses were performed using Libre Office Calc, STATISTICA 6, and IBM SPSS Statistics 23.0. All 29 participants provided a 7-day actigraphy record as well as a single 08:00 am blood sample in winter. The blood for the gene expression was also available: n = 25 (summer), n = 22 (spring), and n = 21 (autumn). To ensure the reliability of the results, we first assessed the normality of each variable’s distribution using the Shapiro–Wilk W-test and Kolmogorov–Smirnov tests. Based on the test outcome, we used either the parametric Student *t*-test, Mann–Whitney U-test, or the non-parametric Kruskal–Wallis H-test to compare the variables across different seasons and populations. Tukey’s Honestly Significant Test (HST) was used for post hoc analysis of between-season differences. Additionally, we applied multivariable ANCOVA with sigma-restricted parameterization to identify the most significant actigraphy-derived predictors while controlling for potential confounding factors, including age, sex, body mass index (BMI), and population. The amplitude (a measure of half of the extent of predictable variation within a 24 h cycle), acrophase (a measure of the time of the overall high values in a cycle), and MESOR (Midline Estimating Statistic of Rhythm, or a rhythm-adjusted mean) values were estimated by cosinor analysis [21].

## 3. Results

NR1D1 expression data were normally distributed in each season (Shapiro–Wilk’s W > 0.935; *p* > 0.115). Season greatly affected REV-ERBα expression, as shown by ANOVA, with the highest expression during the summer solstice and the lowest expression during the winter solstice (F(3,99) = 8.614; *p* = 0.00004) (Figure 1). Tukey’s HST validated REV-ERBα expression to be higher during the summer solstice than the winter solstice (*p* = 0.0002) and the spring equinox (*p* = 0.044), and higher during the autumn equinox than during the winter solstice (*p* = 0.0011). Age, sex, and BMI were not significantly associated with REV-ERBα expression or its seasonal pattern (Table 1). Overall, REV-ERBα expression was significantly higher in native individuals indigenous to the high Arctic latitudes than in newcomers (F(1,101) = 5.769; *p* = 0.018), with both populations showing a similar seasonal pattern of REV-ERBα expression (Figure 2).

REV-ERBα expression was associated with actigraphy-derived measures both across and within seasons. Appendix A show seasonal changes in the MESOR of light exposure and in the MESOR and phase of physical activity. Appendix A displays the correlation matrix documenting the associations between REV-ERBα expression and the variables recorded by actigraphy. Overall measures of daytime light and blue light exposure, such as M, 24 h A, and M10, and measures of physical activity (M, 24 h A, and M10) consistently demonstrated stronger associations with REV-ERBα expression compared to the indices of sleep or wrist temperature. REV-ERBα expression was also significantly associated with the circadian phase of light exposure. Inter-individual differences in NR1D1 expression within distinct seasons were more closely linked to physical activity (lower correlation in winter (r = 0.323, *p* = 0.087) and higher at summer (r = 0.566, *p* = 0.008)) than to light exposure. While light exposure did not correlate with REV-ERBα expression in winter when the light availability was lowest (r = 0.052, *p* = 0.791 for LE MESOR), it correlated in the summer when light was abundant (r = 0.480, *p* = 0.038 after correcting for two outliers). Seasonal changes in REV-ERBα expression were closely coupled with the amount and timing of LE.

To further elucidate the relationship between the timing of light exposure and REV-ERBα expression, NR1D1 expression was correlated with the average light exposure in consecutive 30 min epochs across participants, yielding 48 correlation coefficients. The results revealed time windows when light exposure correlated most strongly with REV-ERBα expression (Figure 3). The strongest correlations were observed in the early morning and even more prominently after noon. A two-component model, incorporating both 24 h and 12 h harmonics, approximated well the correlation pattern (F = 34.4, *p* < 0.00001), validating two significant peaks: a minor morning peak at 07:18 and a major evening peak at 19:31. These findings suggest a complex rhythmic relationship between light exposure and REV-ERBα expression, with distinct morning and evening peaks. Interestingly, a similar approach used to search for the epochs of the strongest correlation between physical activity and REV-ERBα expression only found the 12 h harmonic to be significant (F = 10.5; *p* = 0.0002), accounting for two daily peaks in the morning at 07:26 and in the evening at 19:26, as shown in Appendix A. While the evening peak predicted by the model closely matched the actual peak, the first peak predicted by the model preceded the actual peak observed at 12:30.

Three advanced general regression models using sigma-restricted parameterization examined the factors affecting REV-ERBα expression. To avoid the over-representation of seasons with similar light conditions (spring and autumn), we categorized the data into three distinct seasons: winter, spring, and summer, omitting data from autumn, as shown in Table 1. In the first model, we included the MESOR of light exposure along with age, sex, BMI, and population, as well as the 24 h phase of light exposure. The acrophases of light exposure were in a narrow range with a 95% confidence interval within 60 min (Appendix A), justifying their consideration in the linear analysis. This model revealed that the MESOR and phase of light exposure and population were significant predictors of REV-ERBα expression. The MESOR of light exposure accounted for the largest portion of variability in REV-ERBα expression, achieving an observed power of 0.974. The second model considered the MESOR of physical activity, which emerged as another significant predictor of NR1D1 expression alongside the MESOR and acrophase of light exposure. In the third model, we replaced light exposure with seasonality, given their close relationship. This model reaffirmed that the phase of light exposure and the MESOR of physical activity were significant predictors of NR1D1 expression.

## 4. Discussion

The current study examined seasonal and indigeneity aspects of REV-ERBα (NR1D1) expression in human blood mononuclear cells under the drastically varying seasonal light conditions of the Arctic region. Our findings indicate that REV-ERBα exhibits significant seasonality, with its peak expression during the summer solstice and lowest expression during the winter solstice in both Arctic native and non-native residents, while the overall expression in native Arctic residents is higher than in newcomers, particularly at the summer solstice. Our data also show that daytime light exposure is a strong predictor of REV-ERBα expression, suggesting that environmental light conditions significantly affect this regulatory mechanism. The observed seasonal variation highlights the dynamic nature of REV-ERBα expression, likely reflecting the adaptation to changing photoperiods characteristic of high-latitude environments. Our results add novel human data that are in line with the previous finding showing photoperiodic changes in Rev-erbα expression in hamster tissues with a lower expression and delayed phase in short photoperiods [22,23,24] similar to the ambient light conditions in the Arctic winter [16,17]. It was hypothesized that REV-ERBα may interact with melanopsin to regulate the sensitivity of the rod-mediated intrinsically photosensitive retinal ganglion cell (ipRGC) pathway, thereby coordinating activity in response to ambient blue light exposure [25].

As light exposure only correlated weakly with REV-ERBα expression in some seasons, the results from the three advanced general regression models suggest that the pronounced yearly rhythm in REV-ERBα expression might be driven by the amount and timing of light exposure. A sufficient amount of daylight exposure may be required to boost REV-ERBα expression; alternatively, other factors may account for the variation in morning REV-ERBα expression, including phasic differences in the circadian rhythm of REV-ERBα expression. While individuals vary significantly in their sensitivity to light, there was no clear pattern in the response of melatonin to seasonal changes across individuals. However, seasonal variations in dim light melatonin onset and the 24 h acrophase of melatonin [17] were similar to those in REV-ERBα expression observed herein, suggesting that REV-ERBα expression responds in a similar way as melatonin to seasonal changes, with overall light exposure and its timing likely influencing these fluctuations. Additionally, physical activity appears to contribute to individual differences in the response to seasonal changes in REV-ERBα expression.

Given that REV-ERBα expression in our study was assessed during or soon after the expected peak of its expression [3,6,7], one may suggest that like the circadian awakening response of cortisol, it could be in part a response to “activity-related demands” [26] that also apply to explain REV-ERBα’s specific morning expression. Furthermore, it can be noted that not only is the circadian peak of cortisol close to the peak of REV-ERBα expression in human blood cells, cortisol also peaks during the summer solstice in this very cohort, as previously reported [17].

Overall, seasonal patterns of REV-ERBα expression were similar between native and non-native residents, yet subtle differences in expression levels suggest that long-term adaptation to extreme light conditions may affect the circadian system’s sensitivity to environmental cues. Further exploration of the genetic and epigenetic variations could provide deeper insights into these adaptive mechanisms. Synthetic REV-ERB agonists have been investigated for their potential to synchronize circadian rhythms and modulate metabolic processes, resulting in improved cardiovascular health and reduced obesity through enhanced energy expenditure and metabolic gene expression [27]. Given the established connection between REV-ERBα and metabolic health [9,11], our findings underscore the implications of circadian disruption—often being linked to metabolic disorders—in challenging light environments. Additionally, animal studies indicate that light exposure can enhance the leukocyte immune response, mediated by REV-ERBα. For instance, daytime exposure to blue light significantly improved immune function and survival rates during infections by entraining circadian rhythms, shifting the autonomic balance toward parasympathetic dominance, and activating the REV-ERBα protein [28]. This activation ultimately enhanced pathogen clearance and reduced inflammation, including in the brain [29,30,31]. Thus, REV-ERBα activation may play a crucial role in protecting the organism against neurodegenerative diseases [32,33], cancer [34], and in preventing cardiac [35,36], hepatic [37], and pulmonary fibrosis [38,39].

The results obtained in this study suggest that timed physical activity and light exposure may act in tandem to facilitate REV-ERBα expression that may help to maintain robust circadian rhythmicity and withstand disruptive challenges [40]. Together, these findings highlight the need for further research to clarify the interactions among timed physical activity, seasonal variations in light exposure, and REV-ERBα expression. Understanding these mechanisms is crucial for exploring how optimizing daytime light exposure may help to mitigate the negative effects associated with circadian disruptions, poor light hygiene, and the heightened risk of inflammatory and metabolic disorders.

Study limitations: While our study provides valuable insights into the seasonal and environmental influences on REV-ERBα expression, it is important to acknowledge certain limitations. First, the sample size was relatively small, limiting the generalizability of our findings. Second, the focus on a single time point (08:00) of REV-ERBα determination, which is expected to be close to the average time of its acrophase, precludes a comprehensive understanding of daily oscillations in REV-ERBα expression and how its circadian pattern may differ across seasons beyond its mean value. As this study assessed REV-ERBα expression only during one point in time, both the differences in average 24 h expression or changes in the 24 h amplitude and/or phase that are expected to be advanced in the longer photoperiod of the summer solstice could contribute to the seasonal changes revealed in this study. Participants in this study, however, had a fairly well-synchronized circadian rhythm since the acrophases of light exposure were mostly within a 60 min window (Appendix A). Future studies should incorporate multiple sampling times throughout the day to capture the full circadian profile of REV-ERBα, as we are planning to do in a follow-up study. Additionally, exploring the expression of other key circadian regulators, such as REV-ERBβ, would provide a more complete picture of the circadian machinery’s response to environmental challenges. Adding other clock genes (e.g., BMAL1, PER, CRY) in future studies would also provide a more comprehensive view of seasonal circadian regulation. We opted to study REV-ERBα as a start, considering that it modulates the rhythmic expression of several core clock genes [2,8]. Longitudinal studies tracking individuals over extended periods could also shed light on inter-individual variability and the long-term consequences of circadian misalignment. Our study underscores the importance of considering environmental factors, particularly light exposure, in understanding the dynamics of REV-ERBα expression and its implications for circadian rhythm regulation and metabolic health. These findings lay the groundwork for future investigations aimed at developing targeted interventions to mitigate the negative effects of circadian disruption in extreme environments.

## 5. Conclusions

Overall, our findings reveal a clear seasonal pattern in REV-ERBα expression, with the levels surging to a peak during the summer solstice and dropping to a nadir during the winter solstice, independently of sex, age, or indigeneity that could be closely coupled with seasonal changes in the abundance of daylight. Light received after noon is most closely coupled with REV-ERBα expression. Natives have a higher morning REV-ERBα expression overall and at the summer solstice.

## Figures and Tables

**Figure 1 biology-14-00231-f001:**
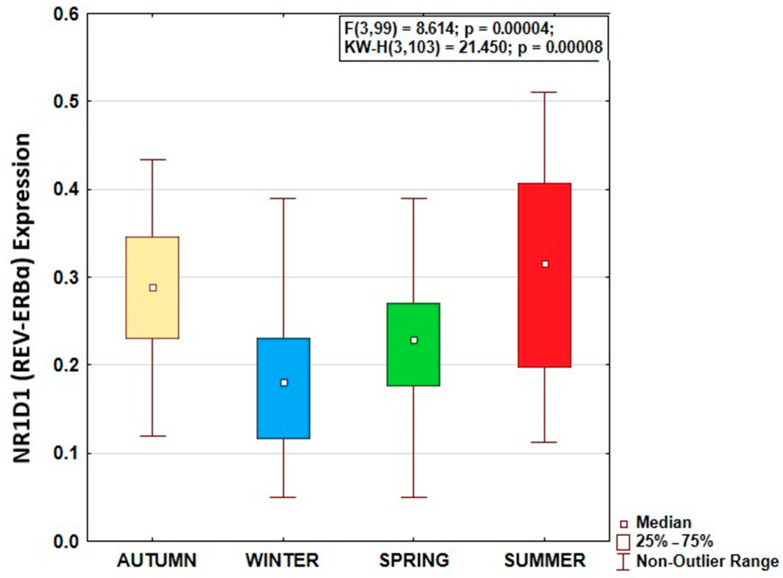
Seasonality of morning (08:00 local time) NR1D1 (REV-ERBα) expression in Arctic residents.

**Figure 2 biology-14-00231-f002:**
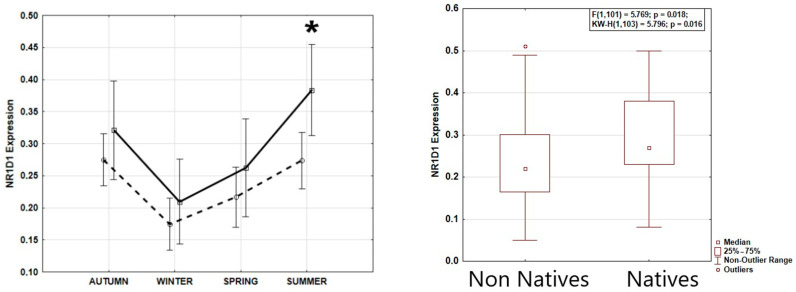
Higher NR1D1 (REV-ERBα) expression in native Arctic population with seasonal pattern similar to that of newcomers. (**Left**): natives—solid line, non-natives—dashed line. ANOVA confirms similar seasonal patterns in both groups, F_(3,95)_ = 0.673, *p* = 0.570. Numerically higher expression during all seasons in natives, and significantly higher in summer (*), F = 6.316, *p* = 0.019. (**Right**): average expression by group.

**Figure 3 biology-14-00231-f003:**
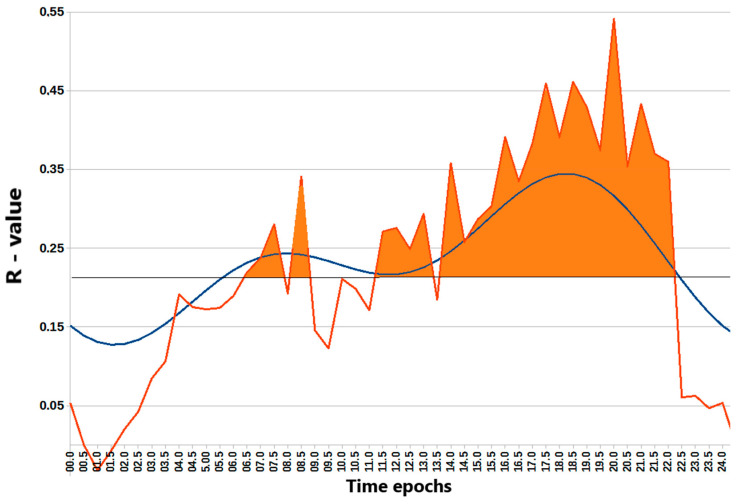
Chart of the r-values from a linear regression of NR1D1 (REV-ERBα) expression with light exposure (ordinate) in consecutive 30 min time epochs (abscissa). The horizontal gray line corresponds to the threshold of significance after Benjamini–Hochberg’s correction for multiple testing at 0.1. Blue curve: cosinor two-component model rejects the zero-amplitude assumption of no rhythmicity; F = 34.4; *p* < 0.00001, and bimodal approximation validates a minor morning peak at 07:18 and a major evening peak at 19:31. The correlation of higher light exposure with higher NR1B1 (REV-ERBα) expression after correction for multiple testing is significant during the time epochs shaded in orange. Note that this figure addresses the question about the prediction of NR1D1 expression based on seasonal data, examining the correlation of expression with the amount of light exposure in different time windows of 30 min each. While it does not examine directly the effect of the duration of light exposure (during a 24 h day) on NR1D1, Figure 1 and Figure 2 indirectly show that the longer daylight of summer is associated with a peak in NR1D1.

**Table 1 biology-14-00231-t001:** Summary of advanced general regression models with sigma-restricted parameterization of factors associated with NR1D1 (REV-ERBα) expression.

Variable	β	β (95% CI)	F	*p*-Value	Partial Eta-Squared	Observed Power
**Model 1**						
**Light MESOR**	**0.437**	**(0.217; 0.657)**	**15.686**	**<0.001**	**0.194**	**0.974**
**Light phase**	**0.299**	**(0.085; 0.513)**	**7.78**	**0.007**	**0.107**	**0.786**
**Population**	**0.241**	**(0.025; 0.456)**	**4.964**	**0.029**	**0.071**	**0.593**
Age	−0.095	(−0.309; 0.119)	0.782	0.38	0.012	0.14
BMI	0.071	(−0.142; 0.283)	0.438	0.511	0.007	0.1
Sex	0.062	(−0.159; 0.284)	0.316	0.576	0.005	0.086
**Model 2**						
**Light MESOR**	**0.348**	**(0.121; 0.575)**	**9.403**	**0.003**	**0.128**	**0.855**
**Light phase**	**0.281**	**(0.073; 0.489)**	**7.294**	**0.009**	**0.102**	**0.758**
**Physical Activity MESOR**	**0.257**	**(0.033; 0.482)**	**5.253**	**0.025**	**0.076**	**0.617**
Population	0.177	(−0.040; 0.392)	2.63	0.11	0.039	0.359
Age	−0.110	(−0.318; 0.097)	1.125	0.293	0.017	0.181
BMI	0.108	(−0.101; 0.317)	1.063	0.306	0.016	0.174
Sex	0.032	(−0.184; 0.249)	0.089	0.767	0.001	0.06
**Model 3**						
**Seasons**	**0.447**	**(0.262; 0.632)**	**23.209**	**<0.0001**	**0.266**	**0.997**
**Physical Activity MESOR**	**0.334**	**(0.140; 0.528)**	**11.786**	**0.001**	**0.156**	**0.922**
**Light phase**	**0.208**	**(0.023; 0.393)**	**5.067**	**0.028**	**0.073**	**0.601**
Population	0.146	(−0.052; 0.344)	2.164	0.146	0.033	0.305
Sex	0.104	(−0.091; 0.298)	1.132	0.291	0.017	0.182
Age	−0.082	(−0.109; 0.273)	0.737	0.394	0.011	0.135
BMI	0.07	(−0.122; 0.262)	0.537	0.466	0.008	0.112

Significant co-factors are in **bold**.

## Data Availability

The data presented in this study are available on reasonable request from the corresponding author. The data are not publicly available due to privacy.

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
