# Peer review of "Light Exposure, Physical Activity, and Indigeneity Modulate Seasonal Variation in NR1D1 (REV-ERBα) Expression"

_biology, 2025, doi:10.3390/biology14030231_

Round 1

Reviewer 1 Report

Comments and Suggestions for Authors

In this study, the authors provides valuable data on the seasonal variation of REV-ERBα expression in humans, an area that has been largely unexplored compared to animal models. The findings may contribute to understanding how environmental factors, such as light exposure and physical activity, influence circadian gene regulation. However, I have the following concerns that the authors should consider and address.

  1. This study used a small Sample Size, with only 29 participants, the sample may not be fully representative, limiting the generalizability of the findings. I recommend increase ther sample size.
  2. REV-ERBα expression was only measured at 08:00 AM, which misses the full circadian profile of its expression throughout the day. I recommend additional time points could help determine whether phase shifts in REV-ERBα occur across seasons.
  3. REV-ERBα is an important circadian regulator, but other clock genes (e.g., BMAL1, PER, CRY) should be examined to provide a more comprehensive view of seasonal circadian regulation.

Author Response

Q1: In this study, the authors provides valuable data on the seasonal variation of REV-ERBα expression in humans, an area that has been largely unexplored compared to animal models. The findings may contribute to understanding how environmental factors, such as light exposure and physical activity, influence circadian gene regulation. However, I have the following concerns that the authors should consider and address.

A1: We are grateful to the reviewer for a thorough evaluation of our work and for raising important questions.

Q2: This study used a small Sample Size, with only 29 participants, the sample may not be fully representative, limiting the generalizability of the findings. I recommend increase ther sample size.

A2: We appreciate the reviewer’s concern regarding the sample size of 29 participants. We acknowledge that a larger sample size would ideally enhance the validity of our findings. However, given the logistical challenges of conducting research in remote Arctic regions (latitude 65°58′ - 66°53 N, longitude 66°60′ - 76°63′ E), and particularly the difficulties associated with recruiting and retaining participants for longitudinal studies involving repeated measures across all four seasons, a larger sample size was not feasible for this project.

It is important to note that this study employs a repeated-measure design, where each participant provided data across all four seasons. This significantly increases the statistical power of the analysis compared to a between-subjects design with the same number of participants. By collecting seven-day actigraphy records and morning blood samples from each participant in each season, we have gathered a rich and detailed dataset, effectively increasing the number of observations.

We clarified the text, line 149 as follows: “All 29 participants provided a 7-day actigraphy record as well as a single 08:00 am blood sample in winter. Blood for gene expression was also available: n=25 (summer), n=22 (spring), and n=21 (autumn)..

Furthermore, we emphasize the unique and relatively understudied population we are investigating: Arctic residents. The challenges inherent in studying this population justify a careful and in-depth analysis of the data we have collected, even with a relatively modest sample size. While we acknowledge that our findings may not be directly applied to different populations, they provide valuable insights into the seasonal physiological and behavioral patterns within this specific Arctic community. In this study, we planned only morning sampling in order to collect data for BodyTime single measurement panel (in accordance with doi:10.1172/JCI120874, a method on determination of circadian time from a single blood sample).

Finally, as the Academic Editor noted, our statistical analyses were deemed adequate for the sample size and the study design, and no significant methodological biases were identified.

We have carefully considered the limitations of our sample size in the interpretation of our results and have clearly stated these limitations in the discussion section of the manuscript. Future research should focus on replicating these findings in larger and more diverse Arctic populations to further validate the generalizability of our observations. Please note that this is addressed in the Study Limitations section.

Q3 REV-ERBα expression was only measured at 08:00 AM, which misses the full circadian profile of its expression throughout the day. I recommend additional time points could help determine whether phase shifts in REV-ERBα occur across seasons.

A3: We thank the Reviewer for raising this important point. We agree that incorporating a 24-hour sampling profile would have significantly enhanced the study’s ability to investigate circadian rhythms. However, the current protocol was designed with a single morning blood sample collection to minimize participant burden, which was crucial for maximizing recruitment and retention in this challenging Arctic research setting. We recognize this as a limitation, but believe the trade-off was necessary to achieve the sample size we obtained for this longitudinal seasonal study as clarified in our answer to Q2. The encouraging results from our morning sampling have indeed motivated us to design a follow-up study specifically focused on characterizing 24-hour expression patterns. We believe this future research will provide a valuable complement to the findings presented in this manuscript.

While this is discussed in the Study Limitations section, we added a statement regarding our intention to sample around the clock in a follow-up study..

Q4: REV-ERBα is an important circadian regulator, but other clock genes (e.g., BMAL1, PER, CRY) should be examined to provide a more comprehensive view of seasonal circadian regulation.

A4: We thank the Reviewer for highlighting the importance of examining other clock genes for a more comprehensive understanding of seasonal circadian regulation. We agree that investigating genes such as BMAL1, PER, and CRY would provide valuable insights, and we, indeed, plan to analyze PER2 and CRY1 expression in future studies to build upon the findings presented herein. We added a sentence to this effect in the Limitations section as follows: “Adding other clock genes (e.g., BMAL1, PER, CRY) in future studies would also provide a more comprehensive view of seasonal circadian regulation. We opted to study REV-ERBα as a start, considering that it modulates the rhythmic expression of several core clock genes [2,8].”

Reviewer 2 Report

Comments and Suggestions for Authors

The circadian clock is an internal molecular oscillator that regulates an organism's physical, mental, and behavioral cycles over a 24-hour period. The circadian clock has many important functions, such as regulating sleep, metabolism, and memory. It also plays a role in cancer and other diseases. NR1D1 is one of the two Rev-Erb proteins in the nuclear receptor (NR) family of intracellular transcription factors. It plays an important role in regulating the core circadian clock by inhibiting the positive clock element Bmal1. The expression of NR1D1 mRNA and protein shows strong circadian oscillations. This study shows that NR1D1 expression in Arctic human monocytes at 8am has a seasonal rhythm and is higher in indigenous than non-indigenous populations. The findings may be of interest to the chronobiology and metabolism communities. The paper is well written and the references are appropriate.

Comments:

Figure 1. How many weeks were tracked per season? NR1D1 mRNA fluctuates across 24 hrs but the samples were all collected at 8 am and therefore are not representative of daily expression of NR1D1.

Table 1. Why were three models used to analyze data? What are the differences between these models?

Figure 2. Another factor to consider is the participant's circadian rhythm. If individuals have different phases or cycle lengths, the timing of sample collection should be adjusted based on their circadian phases or period lengths.

Figure 3. There is a lack of experimental basis for using seasonal data to predict daily NR1D1 expression. Has the prediction been validated? Does NR1D1 expression correlate with light exposure duration?

Author Response

The circadian clock is an internal molecular oscillator that regulates an organism's physical, mental, and behavioral cycles over a 24-hour period. The circadian clock has many important functions, such as regulating sleep, metabolism, and memory. It also plays a role in cancer and other diseases. NR1D1 is one of the two Rev-Erb proteins in the nuclear receptor (NR) family of intracellular transcription factors. It plays an important role in regulating the core circadian clock by inhibiting the positive clock element Bmal1. The expression of NR1D1 mRNA and protein shows strong circadian oscillations. This study shows that NR1D1 expression in Arctic human monocytes at 8am has a seasonal rhythm and is higher in indigenous than non-indigenous populations. The findings may be of interest to the chronobiology and metabolism communities. The paper is well written and the references are appropriate.

We appreciate the Reviewer’s thorough evaluation of our work and are grateful for the encouraging feedback.

Comments:

Q1: Figure 1. How many weeks were tracked per season? NR1D1 mRNA fluctuates across 24 hrs but the samples were all collected at 8 am and therefore are not representative of daily expression of NR1D1.

A1: We thank the Reviewer for questions regarding Figure 1 and the sampling protocol. In response to the question about tracking duration, actigraphy data were collected for seven consecutive days in each season. Regarding the NR1D1 mRNA measurements, we acknowledge the Reviewer’s concern that single morning samples may not fully represent the daily expression pattern of NR1D1. However, the primary aim of this study was to assess seasonal differences in circadian timing using the BodyTime single-measurement panel (doi:10.1172/JCI120874). This panel is specifically designed to estimate internal circadian time from a single blood sample. As we noted in our response to Reviewer 1, the logistical challenges of conducting research in remote Arctic regions, particularly with a repeated-measure design across four seasons, made it difficult to obtain a larger sample size or implement a more intensive sampling protocol, especially when blood sampling is necessary. We believe that the insights gained from this study, despite these limitations, provide valuable information about seasonal circadian patterns in this unique population. Future research should focus on analyzing 24-h mRNA expression in the Arctic population as well as in other locations and populations.

Please note that this is discussed in the Study Limitations section.

We also added the following sentence to the Methods section: “All 29 participants provided a 7-day actigraphy record as well as a single 08:00 am blood sample in winter. Blood for gene expression was also available: n=25 (summer), n=22 (spring), and n=21 (autumn).”

Q2: Table 1. Why were three models used to analyze data? What are the differences between these models?

A2: We thank the Reviewer for this insightful question regarding the modeling approach. We employed three distinct general regression models, as described in Table 1, to investigate factors associated with REV-ERBα expression and to account for the potential confounding effects of correlated variables. Model 1: This initial model examined the relationship between REV-ERBα expression and the MESOR and phase of light exposure, along with demographic variables (age, sex, BMI, and population). The purpose of this model was to assess the direct impact of light exposure on REV-ERBα expression. This model revealed that light exposure, both MESOR and phase, and population were significant predictors of REV-ERBα expression. Model 2: Building upon Model 1, this model incorporated the MESOR of physical activity, which is also closely linked to circadian rhythms. We aimed to determine whether physical activity played an independent role in influencing REV-ERBα expression, beyond the effects of light. Model 2 showed that physical activity significantly increased NR1D1 expression. Model 3: Given the close relationship between light exposure and seasonality, we replaced light exposure variables (MESOR and phase) in Model 1 with a categorical variable representing distinct seasons (winter, spring, and summer, omitting autumn to avoid over-representation of seasons with similar light conditions). This allowed us to explore the direct effect of seasonality on REV-ERBα expression, independently of the specific light levels. This Model showed seasonality impact on NR1D1 expression.

These three models built upon each other to examine the independent and combined influences of different factors, beginning with light exposure (Model 1), then incorporating physical activity (Model 2), and finally examining seasonal differences (Model 3). This strategy allowed us to disentangle the complex relationships among these variables and their effects on REV-ERBα expression, providing a more comprehensive understanding of the factors governing its expression patterns. All three models are in agreement in showing that sex, age and BMI have little, if any influence on NR1D1, whereas light (and its timing) seem to be the major determinant, whether it is represented by the MESOR of light or season. An effect of Population that reaches statistical significance in Model 1 might remain detectable in other models based on a larger sample size, as suggested by results shown in Figure 2.

Q3: Figure 2. Another factor to consider is the participant's circadian rhythm. If individuals have different phases or cycle lengths, the timing of sample collection should be adjusted based on their circadian phases or period lengths.

A3: We thank the Reviewer for this valuable comment. To clarify, while this study focused on the relationship between the amount and timing of light exposure across different seasons and REV-ERBα expression (analyzed using the three models described in our previous response), we have addressed the seasonal changes in circadian phase markers (DLMO, melatonin, wrist temperature, and physical activity) in detail in a recent publication (doi: 10.1111/jpi.70023).

As indicated in the text (lines 212-214), all participants had a fairly well synchronized circadian rhythm since acrophases of light exposure were mostly within a 60-minute window (see Supplemental Table 2). We added a comment to this effect in the Study Limitations section.

We also acknowledge the potential contribution of seasonal phase changes to the observed seasonal differences in REV-ERBα expression. As discussed in the Study Limitations section of the manuscript, “As this study assessed REV-ERBɑ expression only during one point in time, both differences in average 24-h expression or changes in 24-h amplitude and/or phase that is expected to be advanced in the longer photoperiod of the summer solstice, could contribute to seasonal changes revealed in this study. Future studies should incorporate multiple sampling times throughout the day to capture the full circadian profile of REV-ERBα.”

Q4: Figure 3. There is a lack of experimental basis for using seasonal data to predict daily NR1D1 expression. Has the prediction been validated? Does NR1D1 expression correlate with light exposure duration?

A4: We thank the Reviewer for this question. Figure 3 addresses the question about the prediction of NR1D1 expression based on seasonal data, examining the correlation of expression with the amount of light exposure in different time windows of 30 minutes each. It does not examine, however, a direct effect of duration of light exposure on NR1D1, even though, indirectly, the longer daylight of summer is associated with a peak in NR1D1. We added this clarification at the end of the legend of Figure 3.

Figure 3 presents a time series analysis showing correlations between NR1D1 expression and light exposure across a 24-hour period, using data collected across the seasons. We employed a cosinor 2-component model (blue curve) to assess the rhythmicity of NR1D1 expression. This model rejected the zero-amplitude assumption (F = 34.4; p < 0.00001), supporting the existence of a predictable rhythmic interaction. The model validated a bimodal approximation with a minor morning peak at 07:18 and a major evening peak at 19:31.

To address and validate further the correlation with light exposure, Figure 3 also displays r-values from a linear regression analysis, where NR1D1 expression is correlated with light exposure in consecutive 30-minute time epochs. It shows that higher light exposure significantly correlates with higher NR1D1 expression when correcting for multiple testing. This correlation is further supported by Model 1 in Table 1, where light MESOR and light phase are significant predictors of NR1D1 expression.

Overall, our analysis presented in Figure 3 and supported by Model 1 demonstrates a significant relationship between light exposure and the expression of NR1D1. The seasonal aspect of these results is analyzed in detail in Figures 1 and 2 and in associated models where three distinct seasons were considered.

OTHER:

A few minor edits were made throughout the manuscript (as shown in tracked changes).